# Asymmetric 1,4-functionalization of 1,3-enynes via dual photoredox and chromium catalysis

Feng-Hua Zhang[1,2,3], Xiaochong Guo[1,2,3], Xianrong Zeng[1,2] & Zhaobin Wang [1,2] ✉

The merger of photoredox and transition-metal catalysis has evolved as a robust platform in organic synthesis over the past decade. The stereoselective 1,4-functionalization of 1,3-enynes, a prevalent synthon in synthetic chemistry, could afford valuable chiral allene derivatives. However, tremendous efforts have been focused on the ionic reaction pathway. The radical-involved asymmetric 1,4-functionalization of 1,3-enynes remains a prominent challenge. Herein, we describe the asymmetric three-component 1,4-dialkylation of 1,3-enynes via dual photoredox and chromium catalysis to provide chiral allenols. This method features readily available starting materials, broad substrate scope, good functional group compatibility, high regioselectivity, and simultaneous control of axial and central chiralities. Mechanistic studies suggest that this reaction proceeds through a radical-involved redox-neutral pathway.

1,3-Enynes serve as a class of fundamental building blocks with diverse reactivity patterns, including 1,2-, 3,4-, and 1,4-functionalization[1–4]. Particularly, the asymmetric 1,4-functionalization of 1,3-enynes provides quick access to chiral allenes, which not only widely occur in natural products and pharmaceuticals[5,6] but also represent one of the most versatile building blocks for the synthesis of complex molecules[7,8]. Various transition-metal complexes (TM = Pd, Cu, Rh, Sc, etc.) have proved to be able to achieve the asymmetric 1,4-functionalization of 1,3-enynes, involving hydrosilylation[9,10], hydroborylation[11], hydroamination[12], hydrocarbonization[13–15], dicarbonization[16], etc.[17–19]. These transformations generally proceeded via an ionic pathway with allenyl or homoallenyl metal intermediates and mainly formed only one axial chirality (Fig. 1a)[3]. On the other hand, the radical 1,4-functionalization of 1,3-enynes via allenyl or propargylic radicals has attracted much attention recently[20–29], but only limited success has been achieved in their asymmetric versions. In 2020, the Bao and Zhang groups[30], and Liu group[31] independently reported the elegant Cu-catalyzed enantioselective synthesis of chiral allenes via the radical 1,4-dicarbonization of 1,3-enynes. Compared to the ionic pathway, these radical reactions could proceed under mild conditions and afford densely functionalized complexes via a multicomponent manner, which expanded the chemical space for the functionalization of 1,3-enynes. Thus, further exploration of new reaction patterns involving radicals could facilitate efficient access to valuable chiral allenes.

The Nozaki–Hiyama–Kishi reaction[32] is one of the most reliable C–C bond construction approaches with various applications in synthesis chemistry[33–36]. However, conventional NHK reactions are generally limited to reductive processes, and stoichiometric amounts of metal reductants and strong Lewis acids (e.g., chlorosilanes and Schwartz's reagent) must be employed to turn over the chromium catalytic cycle[33]. Recent breakthroughs in dual photoredox and chromium catalysis[37–42] have enabled redox-neutral NHK reactions[43–46]. However, the photocatalytic transformations are limited to asymmetric allylations, reported by the Glorius group[47], Kanai group[48,49]. To the best of our knowledge, the asymmetric radical 1,4-functionalization of 1,3-enynes via merging photoredox and Cr catalysis remains underdeveloped.

As our ongoing efforts in Cr-catalyzed radical-involved reactions[50], we anticipate that the propargyl radical, which is in equilibrium with the allenyl radical, could be captured by a chiral chromium complex, and subsequent nucleophilic addition to the aldehyde affords the enantioenriched products (Fig. 1a, bottom). To achieve this goal, several challenges have to be addressed: (1) the regioselectivity control of 1,4-functionalization versus

[1]Key Laboratory of Precise Synthesis of Functional Molecules of Zhejiang Province, Department of Chemistry, School of Science, Westlake University, Hangzhou 310024 Zhejiang Province, China. [2]Institute of Natural Sciences, Westlake Institute for Advanced Study, Hangzhou 310024 Zhejiang Province, China. [3]These authors contributed equally: Feng-Hua Zhang, Xiaochong Guo. ✉e-mail: wangzhaobin@westlake.edu.cn

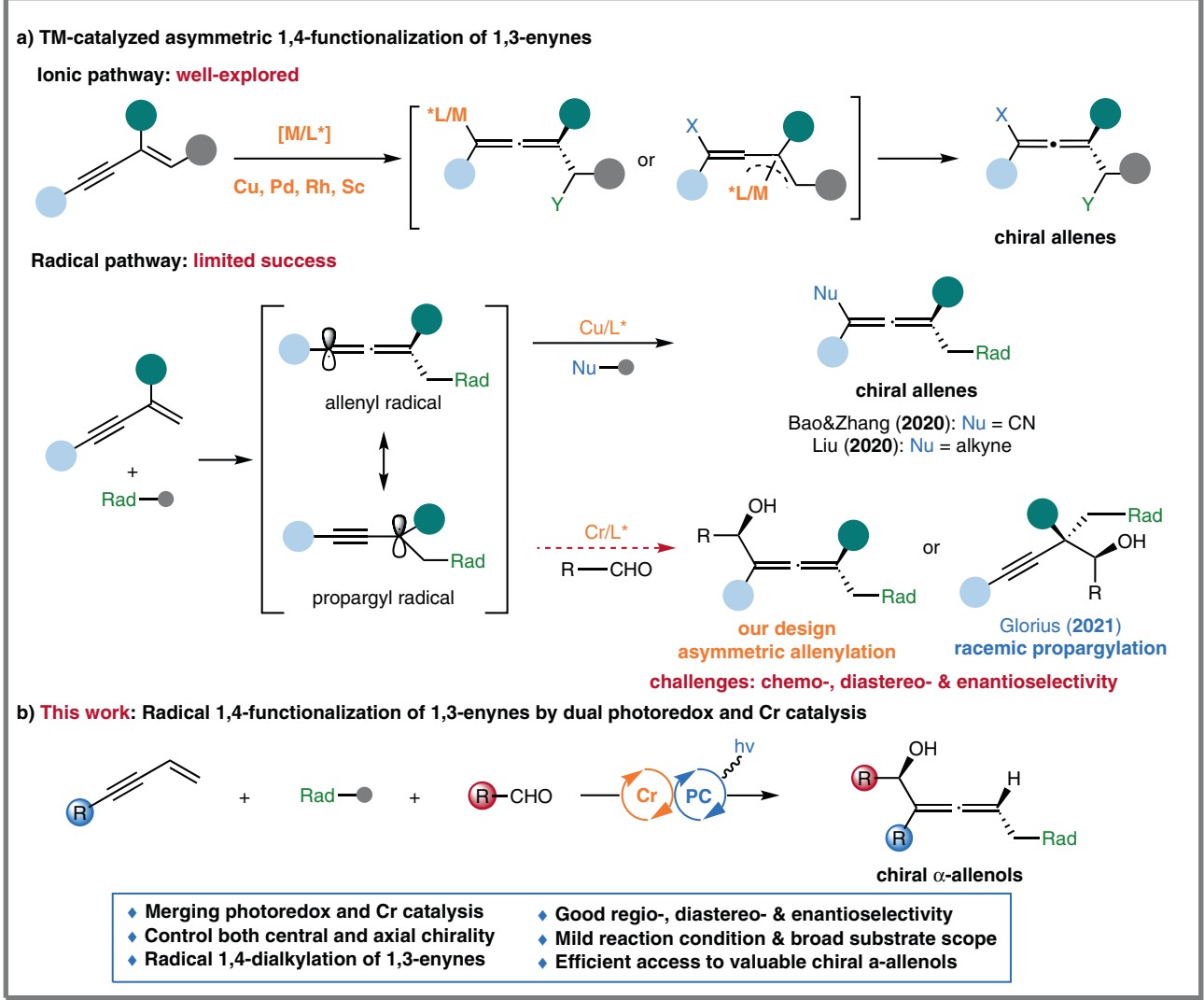

**Fig. 1 | Catalytic asymmetric 1,4-functionalization of 1,3-enynes. a** Transition-metal catalyzed asymmetric 1,4-functionalization of 1,3-enynes. **b** This work: radical 1,4-functionalization of 1,3-enynes by dual photoredox and chromium catalysis. Rad alkyl radical precursors.

1,2-functionalization; (2) the proper choice of radical precursors and photocatalysts to maintain the catalytic cycle; (3) the inhibition of quickly occurring side reactions from reactive radical intermediates or organochromium complexes.

Herein, we describe the three-component asymmetric radical 1,4-functionalization of 1,3-enynes by merging photoredox and chromium catalysis (Fig. 1b). This reaction proceeds efficiently in a redox-neutral manner without an external reductant. And two C–C bonds are simultaneously constructed to provide chiral α-allenols with both a stereogenic center and a stereogenic axis, which serve as essential building blocks in total synthesis (Fig. 2)[51]. Furthermore, the application of versatile and readily accessible materials, including 1,3-enyne, aldehyde, radical precursors, endow the reaction with significant advantages in practical utility.

## Results

### Reaction optimization

With the idea in mind, we initially explored the three-component reaction of benzaldehyde, 1,3-enyne **1**, and DHP ester **2** (Table 1). After detailed investigations of a series of reaction parameters, we determined that the merger of a chiral chromium/cyano-bisoxazoline ((S,R)-**L1**) and a photocatalyst 4-CzIPN could achieve the chemoselective allenylation reaction in good yield and high

diastereoselectivity and enantioselectivity under visible-light irradiation (entry 1). Control experiments establish that CrCl$_2$, 4-CzIPN, and light are critical for this allenylation reaction under these conditions (entries 2–4). A slight decrease in enantioselectivity was detected when using a similar anionic ligand **L2** (entry 5). Other chiral nitrogen-containing ligands are not effective for this reaction under similar conditions (entries 6–10). In the case of **L6**, the homopropargylic alcohol was isolated in 1:3 ratio *vs* the allenol (entry 9). The reaction also performed well in DME, CH$_3$CN, or EtOAc, furnishing the desired chiral allenol only with a slight decrease in yield and dr (entries 11–13). The photocatalyst [Ir(dF(CF$_3$)ppy)$_2$(dtbpy)]PF$_6$ also led to the allenol but with a slight erosion in d.r. and ee (entry 14). Decreasing the catalyst loading to 5 mol% CrCl$_2$ and 6 mol% (S,R)-**L1** led to a drop in yield (entry 15). When increasing the concentration from 0.05 M to 0.1 M, the d.r. decreased from 20:1 to 12:1 (entry 16). And the yield or dr of the allenylation product was only modestly diminished, if 1.2 equivalent of 1,3-enyne **1** and DHP ester **2** are used (entries 17&18). However, adding 1.0 equivalent water to the reaction mixture inhibits the formation of α-allenol **3** (entry 19). The addition of 1.0 mL air to the reaction vessel has a deleterious effect (entry 20). These results indicated that the reaction was sensitive to moisture and air, probably due to the involvement of unstable alkyl chromium complexes.

**Fig. 2 | The importance of chiral allenols.** Bn benzyl, TBS *tert*-butyldimethylsilyl, Me methyl, TIPS triisopropylsilyl, Et ethyl, Tol *p*-methylphenyl, *t*-Bu *tert*-butyl, Boc *tert*-butoxylcarbonyl.

## Substrate scope

We next explored the aldehyde scope (Fig. 3). Gratifyingly, a broad array of aromatic and aliphatic aldehydes serve as effective reaction partners, affording the chiral α-allenols in high yields, good diastereoselectivities, and enantioselectivities (Fig. 3, **3–51**). On a gram scale (1.10 g product), the 1,4-functionalization of 1,3-enyne to product **3** proceeded in 77% yield and 94% ee. A variety of functional groups are compatible with this method, including an aryl halide (e.g., fluoride, chloride, bromide), boronate, methoxy, thioether, amide, carboxylate ester, CF$_3$, furan, thiophene, benzofuran and *N*-alkylated carbazole (Fig. 3, **6–19**). To our delight, heteroaromatic aldehydes, with N, O, or S in the aromatic ring, could react well with the 1,3-enyne and DHP ester under the optimal condition, affording the enantioenriched products efficiently (Fig. 3, **14–23**). Notably, *N*-heteroaromatic rings are widespread in pharmaceuticals and natural products[52]. However, the reactivity of *N*-heteroaromatic aldehydes is rarely demonstrated in previous NHK reactions. As disclosed in recent studies, they generally led to poor yields, including our stereoconvergent allenylation reaction (**20–23**)[50,53–55]. Aliphatic aldehydes, substituted with diverse primary or secondary alkyl chains, participated efficiently in this 1,4-functionalization of 1,3-enynes (**24–39**). However, moderate diastereoselectivities (5:1 dr to 10:1 dr) were generally observed in the cases of primary aliphatic aldehydes (**24–29**), probably resulting from the reduced steric hindrance in comparison with secondary alkyl aldehydes (**30–39**).

Naturally occurring α-amino acids are readily available and act as prevalent feedstocks in asymmetric synthesis[56]. We were delighted to find that the chiral α-amino aldehydes, derived from natural amino acids, served as effective substrates under the standard condition for synthesizing chiral amino alcohols with continuous two stereogenic centers and one chiral axis (Fig. 3, **40–49**). As indicated by the single-crystal structure for products **12** and **42** (see Supplementary Information), the chiral chromium catalyst, rather than existing stereocentres on the chiral aldehydes, predominantly determines the stereochemistry of the allenylation products **40–49**. It is noteworthy that chiral amino alcohols are prevalent synthons in pharmaceuticals and asymmetric catalysis[57]. Finally, the reactivity of α,β-unsaturated aldehydes was tested, and the desired chiral α-allenols were obtained in high yields and diastereoselectivities after increasing the equivalents of 1,3-enyne **1** and DHP ester **2** (Fig. 3, **50** and **51**).

With respect to the DHP esters and 1,3-enynes, the scope of this method is also fairly broad (Fig. 4, **52–69**). For example, moderate to good yields and high diastereo- and enantioselectivities are achieved for the alkyl radical precursors with various alkyl substituents, such as cyclohexyl, oxacyclohexyl, azacyclohexyl, cyclopentyl, cyclopentenyl, and *tert*-butyl (**52–58**). However, using DHP ester with a primary alkyl

substituent furnished the desired allenol **59a** in moderate yield (42%, >20:1 d.r., 85% ee), accompanied by 28% direct alkylation product **59b** in 76% ee. These results indicate that the single electron reduction of the primary alkyl radical by **Cr$^{II}$/L** could compete with its addition to 1,3-enynes. 1,3-Enynes, bearing different acetylenic substituents varying from silyl, alkyl to aryl groups, all reacted smoothly with aryl or alkyl aldehydes and DHP ester **1** to furnish the chiral products efficiently (**60–69**). We found that the use of TMS and TES substituted enynes slightly decreased diastereoselectivity (**61**, **62**), probably due to the variation of steric hindrance. And 1,3-enynes with an aryl group led to the allenols in high enantioselectivity, albeit with moderate regio- and diastereoselectivity (**66–69**). However, the current optimal condition does apply to 1,3-enynes bearing substituents on the C=C bond (Fig. 4, bottom). The use of triisopropyl(3-methylbut-3-en-1-yn-1-yl) silane gave the propargylation product **70** predominantly with poor diastereoselectivity.

Organotrifluoroborates, featuring tetracoordinate boron with strong boron-fluoride bonds, are generally stable toward numerous regents that are often problematic for other trivalent organoborons, and thus have been widely used in Suzuki-Miyaura couplings[58]. Moreover, organotrifluoroborates also prove to be suitable radical precursors for C–C bond construction via photoredox catalysis[59,60]. In this context, we applied them as radical precursors to our newly developed method. After further evaluation of different reaction parameters, we determined an optimal condition with the acridine tetrafluoroborate (PC-2) as the photocatalyst and 2,6-dimethylpyridine hydrochloride as the dissociation reagent. Thus, the representative secondary organotrifluoroborates engaged well in the 1,4-functionalization of enynes with aryl and aliphatic aldehydes to efficiently afford the desired coupling products (Fig. 5a, **53**, **56**, **71**, and **72**). *N*-(Acyloxy)phthalimides (NHPI esters) are widely available from carboxylic acids, and have proved to be priviliged alkyl radical precursors in decarboxylitive cross-couplings[61,62]. Gratifyingly, NHPI esters also work well under a slightly modified condition with Hantzsch ester as the reductant, furnishing the desired allenols in moderate to good yield and high stereoselectivity (Fig. 5b, **1**, **31**, **53**, and **58**).

## Synthetic application

Product transformations were performed to demonstrate the synthetic utility of our newly developed method (Fig. 5c). Chiral α-allenols serve as suitable building blocks in the synthesis of enantioenriched dihydrofurans[63]. The desilylation reaction of **63** proceeded smoothly, affording the chiral α-allenol **73** without losing diastereomeric or enantiomeric excess. The stereoselective electrophilic cyclization of **73** furnished 2,5-dihydrofurans **74** and **75** with good efficiency in axial-to-central chirality transfer (Fig. 5c).

**Table 1 | Effect of reaction parameters**

| Entry | Variation from "standard conditions" | Yield[a] [%] | dr[b] | ee[c] [%] |
|---|---|---|---|---|
| 1 | None | >95 | 20:1 | 94 |
| 2 | Without CrCl$_2$ | <2 | – | – |
| 3 | Without 4-CzIPN | <2 | – | – |
| 4 | Without blue LED | <2 | – | – |
| 5 | L2, instead of (S,R)-L1 | >95 | 20:1 | 92 |
| 6 | L3, instead of (S,R)-L1 | 84 | 3.1:1 | 47 |
| 7 | L4, instead of (S,R)-L1 | 68 | 1.6:1 | 19 |
| 8 | L5, instead of (S,R)-L1 | 52 | 5.5:1 | 34 |
| 9[d] | L6, instead of (S,R)-L1 | 71 | 1.8:1 | 46 |
| 10 | L7, instead of (S,R)-L1 | 79 | 2.8:1 | 93 |
| 11 | DME, instead of THF | 72 | 10:1 | 94 |
| 12 | MeCN, instead of THF | 85 | 20:1 | 95 |
| 13 | EtOAc, instead of THF | >95 | 20:1 | 94 |
| 14 | [Ir(dF(CF$_3$)ppy)$_2$(dtbpy)]PF$_6$ instead of 4-CzIPN | >95 | 11:1 | 88 |
| 15 | 5 mol% CrCl$_2$, 6 mol% (S,R)-L1 | 74 | 20:1 | 94 |
| 16 | 0.1 M, instead of 0.05 M, in THF | >95 | 12:1 | 93 |
| 17 | 1.2, instead of 1.5, equiv 1 and 2 | 80 | 20:1 | 94 |
| 18 | 1.2, instead of 1.5, equiv 1 | 90 | 18:1 | 93 |
| 19 | 1.0 equiv H$_2$O was added | <2 | – | – |
| 20 | 1 mL air (added via syringe) | 55 | 14:1 | 87 |

[a]Yields were determined via [1]H NMR analysis with 1,3,5-trimethoxybenzene as the internal standard.

[b]Drs were determined via [1]H NMR analysis of the crude product.

[c]Ee was determined via HPLC analysis.

[d]Homopropargylic alcohol by-product was observed in 1:3 ratio vs the allenol 3.

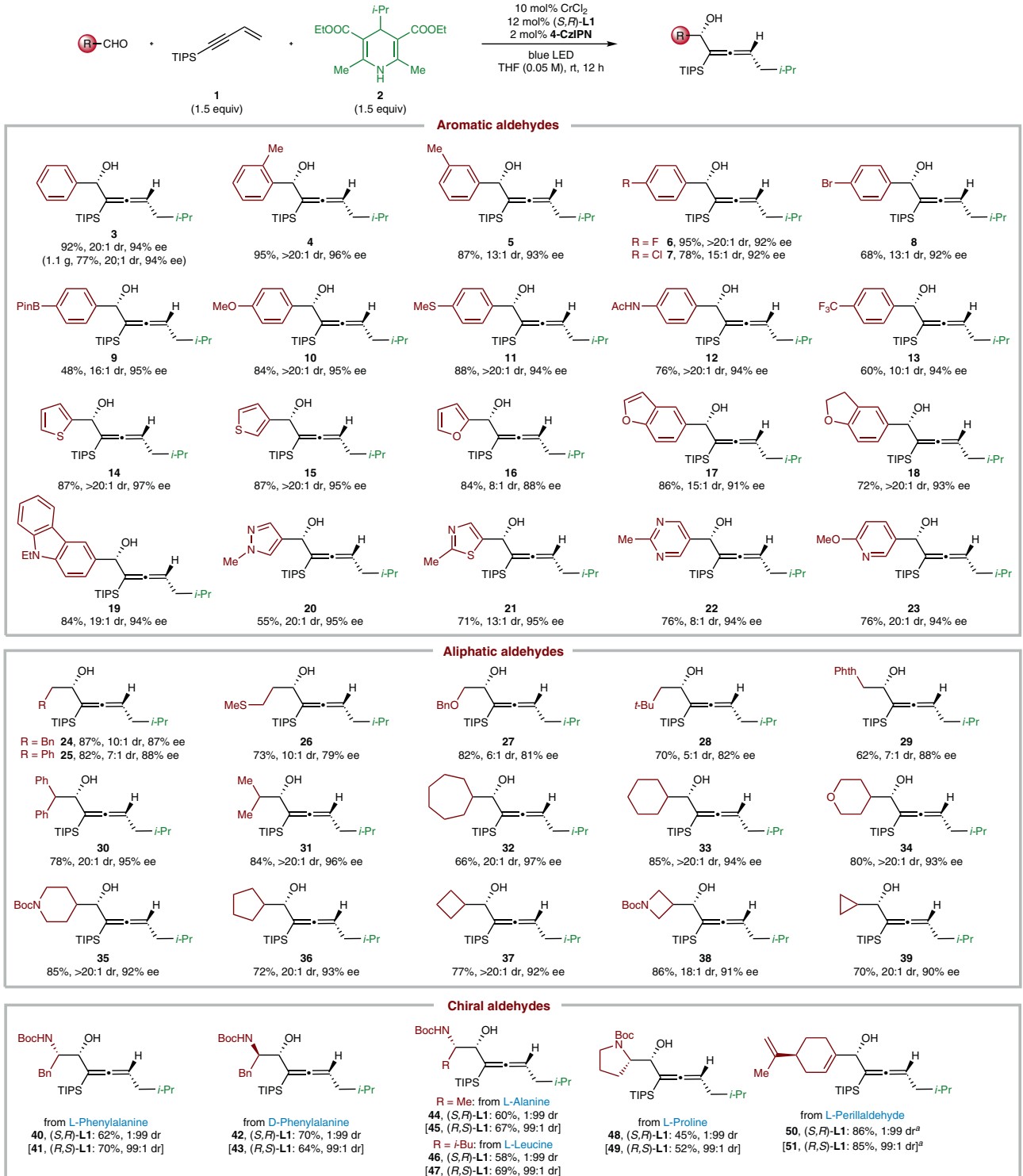

**Fig. 3 | The scope of aldehydes in the 1,4-functionalization of 1,3-enynes.** [a]2.0 equiv of DHP ester and 2.0 equiv of 1,3-enyne were used. *i*-Pr isopropyl, Bpin boronic acid pinacol ester, Ac acetyl.

## Mechanistic observations

A series of conventional experiments were conducted to provide insights into the reaction mechanism (Fig. 6a–c). The addition of 2 equiv of an allyl sulfone under the standard condition led to an adduct **76** in 42% yield, with a trace amount of desired product **3**, which suggested that the reaction might involve the formation of cyclohexyl radical from the DHP ester (Fig. 6a). According to a reported method[64], the quantum yield of this model reaction was

determined to be 0.35. Moreover, the direct correlation between photolysis and product formation is demonstrated by an interval light-dark reaction (Fig. 6b). These results indicate that the radical 1,4-functionalization process undergoes a photoredox, instead of a radical-chain, pathway. As shown in Fig. 6c, the Stern–Volmer luminescence quenching studies proved that the DHP ester, rather than the 1,3-enyne, quenches the excited-state photocatalyst 4-CzIPN, suggesting a reductive quenching pathway.

**Fig. 4 | The scope of DHP esters and 1,3-enynes in the 1,4-functionalization of 1,3-enynes.** [a]4-(*tert*-butyl)-2,6-dimethyl-1,4-dihydropyridine-3,5-dicarbonitrile was used as radical precursor. [b]3.0 equiv of corresponding 1,3-enyne were used. [c]The yield is for allenol product, and the minor regioisomer refers to the propargylation product from the 1,2-functionalization. TMS trimethylsilyl, TES triethylsilyl.

According to our observations and previous reports[44,46,47], a putative mechanism is proposed in Fig. 6d with the model reaction as an example. The excited-state photocatalyst PC* 4-CzIPN* ($E_{1/2}$(*PC/PC$^{\cdot-}$) = 1.35 V vs. SCE in MeCN)[65] is reductively quenched by the DHP ester **2** ($E_{1/2}$ = 1.10 V vs. SCE in MeCN)[66], generating the reduced photocatalyst PC$^{\cdot-}$ and the radical cation **A**. The rapid fragmentation of intermediate **A** affords the isopropyl radical and the pyridinium **B**. The isopropyl radical could either reversibly add to the low valent Cr$^{II}$/**L1** to

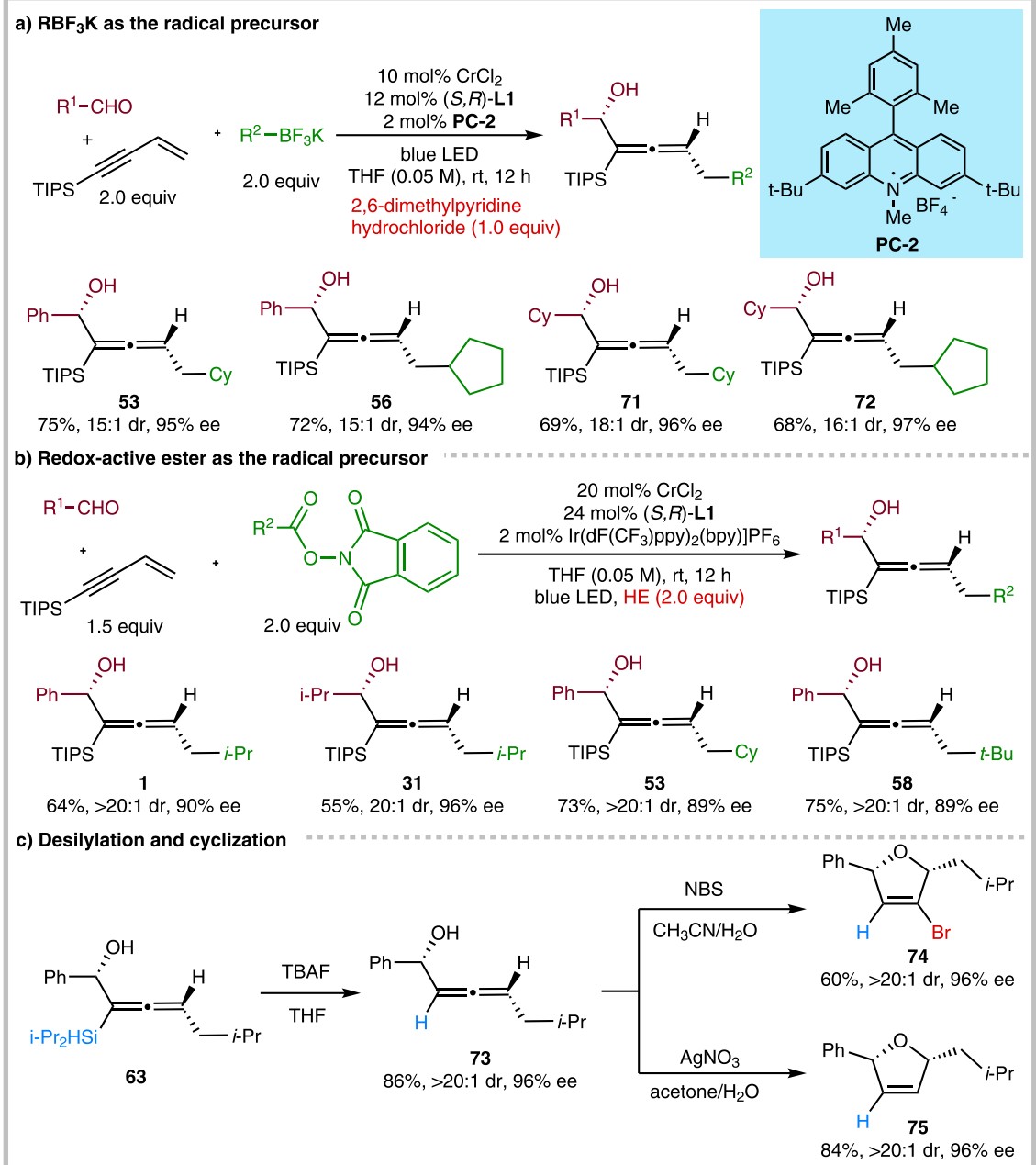

**Fig. 5 | Exploration of other radical precursors and representative synthetic applications. a** RBF$_3$K as the radical precursor. **b** Redox-active ester as the radical precursor. **c** Desilylation and cyclization. NBS *N*-bromosuccinimide.

generate an off-cycle alkyl Cr$^{III}$/**L1** complex[67], or add to the terminus of 1,3-enyne **1** to forge the propargyl radical **C**, which is in equilibrium with the allenyl radical **C'**. The radical capture by Cr$^{II}$/**L1** leads to two equilibrated species, the propargyl chromium **D** and allenyl chromium **D'**. Subsequent nucleophilic attack to benzaldehyde is proposed via a six-member cyclic manner[68], affording intermediate **E**. We believe that the isomerization between intermediates **D** and **D'** is faster than the subsequent nucleophilic addition to aldehydes. So the regioselectivity might be determined in the nucleophilic carbonyl addition step via a possible Zimmerman-Traxler transition state. As observed in the scope study, the steric hindrance of the acetylenic substituents of 1,3-enynes is critical for the high regioselectivity, which favors the allenylation product formation from the propargyl Cr **D**, instead of the allenyl **D'**. The dissociation of the O–Cr bond in **E** by the pyridinium **B**, provides chiral allenol **3**. Finally, the Cr$^{III}$/**L1** is reduced to Cr$^{II}$/**L1** (E$_{1/2}$ = −0.65 V

vs. SCE in H$_2$O, E$_{1/2}$ = −0.51 V vs. SCE in DMF)[47] by the reduced photocatalyst PC$^{•−}$ (E$_{1/2}$ (PC/PC$^{•−}$) = −1.21 V vs. SCE in MeCN) [65], which closes the catalytic cycle.

In conclusion, we described a three-component asymmetric radical 1,4-functionalization of 1,3-enynes via dual photoredox and chromium catalysis. The key to success is using DHP esters under photoredox conditions, thus obviating stoichiometric amounts of metal reductants and dissociation reagents in conventional catalytic NHK reactions. The present method exhibits broad substrate scope with good functional group compatibility, providing efficient access to valuable chiral α-allenols from the readily available starting material. Given the importance of allenols and the growing interest in metalla-photoredox catalysis[37], we anticipate that our protocol will find broad utility in organic synthesis and facilitate the current endeavors to develop dual catalytic systems.

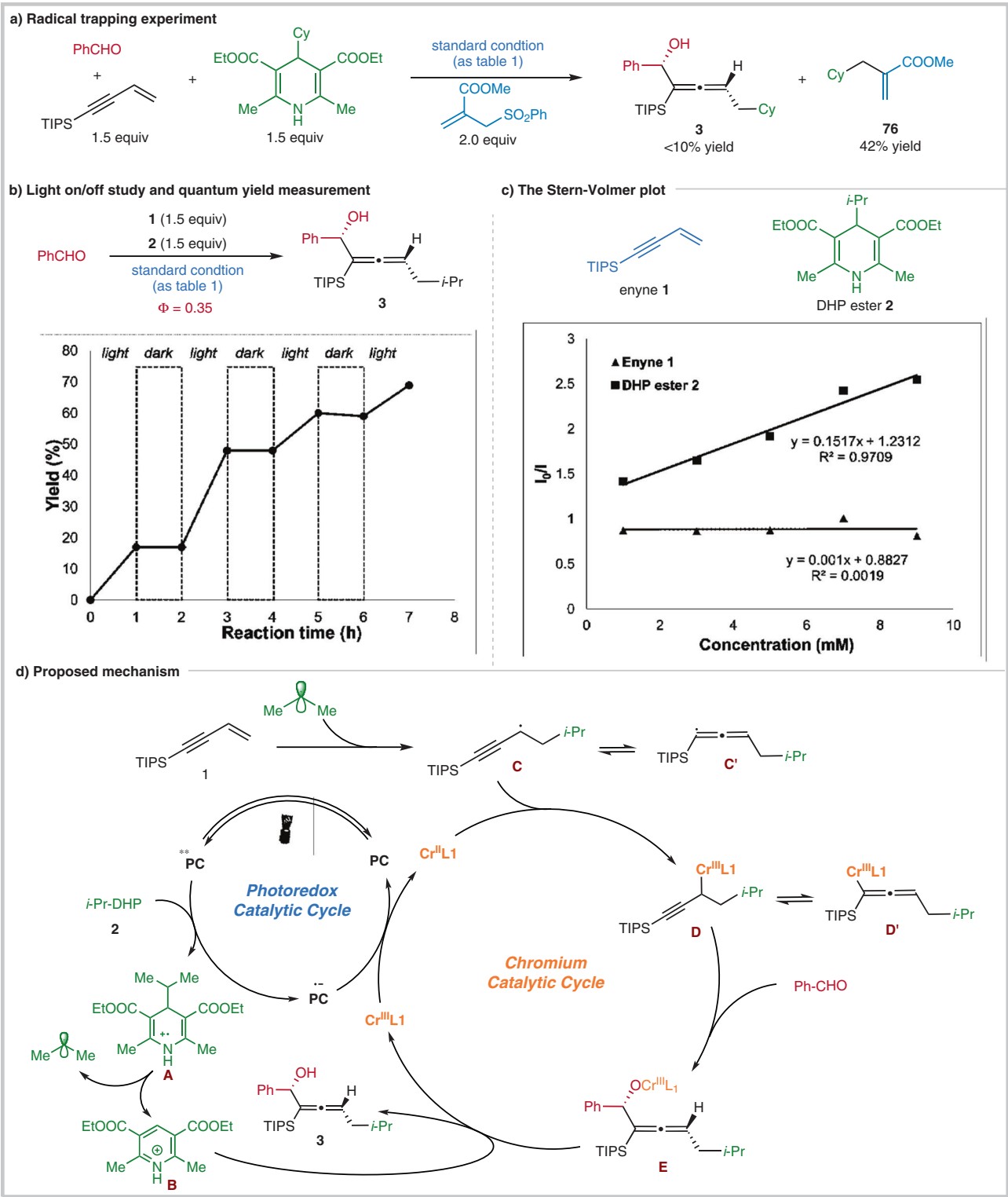

**Fig. 6 | Mechanistic investigations. a** Radical trapping experiment. **b** Light on/off and quantum yield mearsurement. **c** The Stern–Volmer plot. **d** Proposed mechanism for the 1,4-functionalizaton of 1,3-enynes.

## Methods

### General procedure for radical 1,4-functionalization of 1,3-enynes with aldehydes and DHP esters

In a nitrogen-filled glovebox, an oven-dried 20 mL vial with a magnetic stir bar, were charged with the CrCl₂ (5.0 mg, 0.04 mmol, 10 mol%) and (S,R)-**L1** (23.2 mg, 0.048 mmol, 12 mol %). Then 8.0 mL THF was added via syringe. The vial was closed with a PTFE septum cap and then stirred at room temperature for

2 hours. Next, to the prepared catalyst solution were added the 1,3-enynes (0.6 mmol, 1.5 equiv), the aldehydes (0.4 mmol, 1.0 equiv), the DHP esters (0.6 mmol, 1.5 equiv), and photocatalyst 4-CzIPN (6.4 mg, 0.008 mmol, 2 mol%) sequentially. Then the vial was closed with a PTFE septum cap and taken out of the glovebox. The reaction was irradiated with two 20 W 160-440 nm LED for 12 h (tube 5 cm away from lights, fans for cooling, 30–35 °C). After that, the reaction mixture was concentrated and run through a

short silica gel pad with hexanes/EtOAc (3:1) as the eluent. Then the solvent was removed under the reduced pressure. The diastereoselectivity was determined via [1]H NMR analysis of the crude reaction mixture. The residue was purified by flash chromatography to provide the desired product, and the ee was determined via HPLC/SFC analysis.

## Data availability

The data relating to the materials and methods, experimental procedures, HPLC/SFC spectra, mechanism research, and NMR spectra are available in the Supplementary Information. The crystallographic data for compounds **12** and **42** are available free of charge from the CCDC under reference numbers 2130059 and 2130062. All other data are available from the authors upon request.

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

## Acknowledgements

We are grateful for financial support from the National Natural Science Foundation of China (22171231, Z.W.), the China Postdoctoral Science Foundation (2021M692879, F.-H.Z.), and Zhejiang Leading Innovative and Entrepreneur Team Introduction Program (2020R01004). We thank Instrumentation and Service Center for Molecular Science and Physical Sciences at Westlake University for the assistance work in measurement/data interpretation. We thank Dr. Xiaohuo Shi, Dr. Yinjuan Chen, Dr. Zhong Chen, and Dr. Danyu Gu from Instrumentation and Service Center for Molecular Sciences at Westlake University for assistance in mechanistic study.

## Author contributions

F.-H.Z. and X.G. contributed equally to this work. F.-H.Z. and Z.W. conceived this work; F.-H.Z., X.G., and X.Z. designed and conducted all the experiments; F.-H.Z. and Z.W. wrote the manuscript.

## Competing interests

The authors declare no competing interest.
