## [Peer Review File · Nature Communications]

REVIEWER COMMENTS

Reviewer #1 (Remarks to the Author):

Recently, the merger of photoredox and transition-metal catalysis has been proved to be a powerful strategy in organic synthesis. In their previous work (ACIE, 2022, 61, e202117114), Wang and co-workers reported Cr-catalyzed asymmetric synthesis of chiral allenols via catalytic enantioconvergent allenylation of aldehydes with propargyl halides. In the present manuscript, the authors reported a different strategy for the synthesis of chiral allenols via asymmetric three-component 1,4-dialkylation of 1,3-enynes with dual photoredox and chromium catalysis. This protocol featured broad substrate scope, high regioselectivity and simultaneous control of axial and central chiralities. In addition, a series of conventional experiments were conducted for the mechanism study. The present work represents a significant progress in the development of asymmetric radical transformations for the synthesis of valuable chiral products via dual photoredox/transition-metal catalysis. Therefore, this referee recommends that this manuscript could be published in Nature Communications after addressing the following points.

1. Several references involving NHC-catalyzed 1,4-bifunctionalization of 1,3-enynes for facile synthesis of tetra-substituted allenes via radical relay pathway (ACS Catal. 2021, 11, 13363-13373; J. Org. Chem. 2022, 87, 5229-5241.) need to be cited properly. Additionally, the 1,4-sulfonylation of 1,3-enynes via nickel/photoredox dual catalysis was also developed by Lu's group (Chem. Sci., 2021, 12, 13564-13571.) and Wang's group (Org. Chem. Front., 2022, 9, 788-794.).

2. In 2021, Glorius and co-workers reported a radical carbonyl propargylation of 1,3-enynes via dual photoredox/chromium(III) catalysis (ACIE, 2021, 60, 2464). A library of homopropargylic alcohols were accessed from 1,3-enynes, aldehydes and Hantzsch esters. The catalytic system looks similar to this work but resulting in different products. Whether the homopropargylic alcohol products were also detected in the present work? If not, can the authors explain the main reason for this different chemoselectivity? And are there any other side products detected in the present catalytic system?

3. The reaction scope needs to be further examined. In this work, 1,3-enynes bearing substituents on the internal carbon atom of the triple bond were used in the present dual-catalytic system. However, we did not find any examples of 1,3-enynes bearing substituents on the C=C bond.

4. The authors mentioned that "From a practical point of view, it is noteworthy that the yield and dr of the allenylation product are only modestly diminished, if 5 mol% CrCl₂ and 6 mol% (S,R)-L1 are used, the concentration is increased from 0.05 M to 0.1 M, or 1.2 equivalent of 1,3-enyne 1 and DHP ester 2 are used (entries 10-13)." Actually, when decreasing the loading of the catalyst and the ligand, the yield of the desired products decreased by 20%. And when increasing the concentration, the decrease of the dr might not be ignored for practically enantioselective synthesis of the desired products, although the yield was not affected. Thus, this part might be reorganized.

5. In lines 127-130, two "notably" leading sentences were used in succession. It's better to avoid the same sentence pattern.

6. In Scheme 1, the configuration of the α -carbons of the nitrogen atom were not assigned for products 40-43.
7. For the alkyl radical precursors, only secondary alkyl radicals were tolerated with the present system. What about primary and tertiary alkyl radicals? Besides organotrifluoroborates, did the authors evaluate other commonly used alkyl radical precursors, like NIPH esters, Katritzky salts?
8. Since the obtained allenol structure is frequently found in natural products, this referee highly recommends the authors to give one example applied in the synthesis of natural product molecules, so as to further enhance the quality of this work.
9. For the reaction mechanism, the authors mentioned that "As observed in the scope study, the steric hindrance of the acetylenic substituents of 1,3-enynes might be critical for the high chemoselectivity, which favors the allenylation product formation from the propargyl intermediate D, instead of the allenyl D'." If so, is there any evidence to support that the present system is more favorable for the formation of propargyl chromium D? And why the direct nucleophilic attack of D to benzaldehyde was not favorable and observed? In addition, is there any theoretical calculation to support this result? Furthermore, please explain how the diastereoselectivity and enantioselectivity of the newly formed vicinal stereocenters was controlled for this transformation.
10. The HRMS data for (Scheme 1, entry 6) and (Scheme 2, entry 63) need to be retested.
11. Please provide the HRMS data for (Scheme 1, entry 17).

Reviewer #2 (Remarks to the Author):

In this manuscript, Wang and co-workers describe a photoredox/Cr catalyzed three-component regio-, diastereo-, and enantioselective radical 1,4-functionalization of 1,3-enynes, providing efficient access to valuable chiral α -allenols. A reasonable mechanistic cycle and rationale was proposed. The results reported in this paper are both significant and intriguing. This paper is well written in a concise way as a full paper. The supporting information is also in a good shape. Overall,

this synthetic aspect, combined with the mechanistic novelty of this reaction, warrants publication of this interesting work in Nature Communications.

A few comments:

1) The closest precedent of this work is a radical carbonyl propargylation by Glorius et al. (ref. 41), which uses the same catalyst system but in racemic fashion. Considering the relevance of these works, it should be highlighted in Figure 1.

2) The statement “the reaction was sensitive to moisture and air, probably due to the involvement of highly reactive radical intermediates” in page 5 is not accurate. Normally, radical intermediates are not sensitive to moisture and organochromium species is. Therefore, this statement should be modified.

3) The substrate scope is somewhat limited. As for the scope of alkyl-DHPs (Scheme 2), only six alkyl-DHPs have been investigated. To understand the generality and utility of this reaction system, the authors should add more examples of alkyl-DHPs.

(a) The scope does not show any primary or tertiary alkyl-DHPs. Were they not considered or are they not compatible in this reaction?

(b) Could secondary alkyl-DHPs bearing unsymmetric alkyl group be converted to the corresponding products diastereoselectively?

4) Only a few variations from standard conditions are shown in Table 1. As a research paper, reaction condition optimization details should be given in the manuscript or SI, which can help readers better understand the reaction.

5) In Scheme 2, some *rr* values are given for substrates 65-67. What is the other regioisomer? 1,2-functionalization? This information should be given as a footnote in Scheme 2.

6) ¹⁹F-NMR spectra of 6 and 13 should be provided.

Reviewer #3 (Remarks to the Author):

Wang and coworkers reported an asymmetric 1,4 dialkylation of 1,3 enynes via dual photoredox and chromium catalysis, which provides chiral allenols in excellent yields, with good regio-, diastereo- and enantioselectivity. This method also features broad substrate scope, good functional group compatibility. Mechanistic studies have been well studied and could support the proposed mechanism. As a key step, the allenyl radicals or propargylic radicals was enantioselectively trapped by chiral Cr species, then reacts with aldehydes to afford the desired addition product, which is quite similar to the author's report in the recent enantioconvergent coupling of aldehydes with propargyl

halides (as shown in ref 45). In this work, the formation carbon radicals were derived from alkyl radical addition to enynes, where initial radicals were generated from DHP ester 2 under photocatalysis. Notably, this process has been well studied, which reduced the novelty of this manuscript. Overall, this referee believes that the current manuscript is on the board line, and leave the decision to editor to publication in Nature Communication.

Comments:

- (1) The first challenge in line 66 should be regioselectivity control of 1,4-functionalization versus 1,2-functionalization, rather than chemoselectivity control. And some challenges should be refined on the basis of ref 45.
- (2) Please provide the ee values in entries 6 and 7, which may provide more information for readers to know the stereo control with L3 and L4.
- (3) The reason why the reaction was sensitive to moisture, is not attribute to radical intermediate.
- (4) In the proposed mechanism, the authors indicated that the isopropyl radical could either reversibly add to the low valent CrII/L1 to generate an off-cycle alkyl CrIII/L1complex. Did the authors detect any side product? Did the off-cycle reaction affect the titled reaction? How to control the off-cycle reaction?

Point-by-point Response to the Reviewer Comments

Reviewer 1:

“... The present work represents a significant progress in the development of asymmetric radical transformations for the synthesis of valuable chiral products via dual photoredox/transition-metal catalysis. Therefore, this referee recommends that this manuscript could be published in Nature Communications after addressing the following points.”

Comment 1: Several references involving NHC-catalyzed 1,4-bifunctionalization of 1,3-enynes for facile synthesis of tetra-substituted allenes via radical relay pathway (*ACS Catal.* **2021**, *11*, 13363-13373; *J. Org. Chem.* **2022**, *87*, 5229-5241.) need to be cited properly. Additionally, the 1,4-sulfonylarylation of 1,3-enynes via nickel/photoredox dual catalysis was also developed by Lu's group (*Chem. Sci.*, **2021**, *12*, 13564-13571.) and Wang's group (*Org. Chem. Front.*, **2022**, *9*, 788-794.).

Response: We highly appreciate the suggestions from the referee. All these four references were cited as refs 26–29.

Comment 2: In 2021, Glorius and co-workers reported a radical carbonyl propargylation of 1,3-enynes via dual photoredox/chromium(III) catalysis (*ACIE*, **2021**, *60*, 2464). A library of homopropargylic alcohols were accessed from 1,3-enynes, aldehydes and Hantzsch esters. The catalytic system looks similar to this work but resulting in different products. Whether the homopropargylic alcohol products were also detected in the present work? If not, can the authors explain the main reason for this different chemoselectivity? And are there any other side products detected in the present catalytic system?

Response: We thank the referee for these suggestions.

The homopropargylic alcohols were observed under certain conditions. In entry 9 of Table 1, the use of L6 provided the homopropargylic alcohol in 1:3 ratio *vs* the allenol. In Scheme 2, when using aryl 1,3-enynes, the homopropargylic alcohols were isolated as the minor products in poor d.r. (about 1:2). So, the regioselectivity is mainly controlled by the steric hindrance of the terminal substituent on 1,3-enynes. And the chiral ligand also matters.

Accordingly, a footnote was added to Table 1 to provide the reader with

more information about the regioselectivity issue. ^d Homopropargylic alcohol by-product was observed in 1:3 ratio vs. the allenol **1**."

The common by-product detected in most cases is an allene, generated from the sequence of radical addition and protonation (e.g., protonation of Int D' in Figure 4d), or hydrogen atom abstraction of the allenyl radical Int C'.

Comment 3: The reaction scope needs to be further examined. In this work, 1,3-enynes bearing substituents on the internal carbon atom of the triple bond were used in the present dual-catalytic system. However, we did not find any examples of 1,3-enynes bearing substituents on the C=C bond.

Response: We thank the referee for the valuable suggestion. As shown following, the use of triisopropyl(3-methylbut-3-en-1-yn-1-yl)silane under the standard condition in Scheme 2, gave the propargylation product **70** predominantly with poor diastereoselectivity (eq. 1). This might be due to the increasing steric hindrance in the propargylic carbon, favoring the allenyl Cr complex in the proposed Zimmerman-Traxler transition state for nucleophilic addition to aldehydes. However, the use of triisopropyl(pent-3-en-1-yn-1-yl)silane did not provide any desired coupling product, possibly resulted from the decreased reaction rate for alkyl radical addition to the internal 1,3-enyne (eq. 2).

To provide the readers with more information, we have included these results in Scheme 2, the bottom part. And the following statement was added to the main text:

"However, the current optimal condition does apply to 1,3-enynes bearing substituents on the C=C bond (Scheme 2, bottom). The use of triisopropyl(3-methylbut-3-en-1-yn-1-yl)silane gave the propargylation product **70** predominantly with poor diastereoselectivity."

Comment 4: The authors mentioned that "From a practical point of view, it is noteworthy that the yield and dr of the allenylation product are only modestly

diminished, if 5 mol% CrCl₂ and 6 mol% (*S,R*)-L1 are used, the concentration is increased from 0.05 M to 0.1 M, or 1.2 equivalent of 1,3-enyne 1 and DHP ester 2 are used (entries 10-13).” Actually, when decreasing the loading of the catalyst and the ligand, the yield of the desired products decreased by 20%. And when increasing the concentration, the decrease of the dr might not be ignored for practically enantioselective synthesis of the desired products, although the yield was not affected. Thus, this part might be reorganized.

Response: We really appreciate the suggestion from the referee. We agree that our statement is not appropriate. Accordingly, we have modified the main text part as follows:

“Decreasing the catalyst loading to 5 mol% CrCl₂ and 6 mol% (*S,R*)-L1 led to a drop in yield (entry 15). When increasing the concentration from 0.05 M to 0.1 M, the d.r. decreased from 20:1 to 12:1 (entry 16). And the yield or dr of the allenylation product was only modestly diminished if 1.2 equivalent of 1,3-enyne 1 and DHP ester 2 are used (entries 17&18).”

Comment 5: In lines 127-130, two “notably” leading sentences were used in succession. It's better to avoid the same sentence pattern.

Response: We thank the referee for this good suggestion. The second “Notably” was changed to “However”.

Comment 6: In Scheme 1, the configuration of the α -carbons of the nitrogen atom were not assigned for products 40-43.

Response: We sincerely appreciate this suggestion. Accordingly, we have reorganized Scheme 1 and assigned the absolute configuration of the α -carbons of the nitrogen atom for products 40–43.

Comment 7: For the alkyl radical precursors, only secondary alkyl radicals were tolerated with the present system. What about primary and tertiary alkyl radicals? Besides organotrifluoroborates, did the authors evaluate other commonly used alkyl radical precursors, like NHPI esters, Katritzky salts?

Response: We really appreciate the helpful suggestions from the referee.

The use of 4-(*tert*-butyl)-2,6-dimethyl-1,4-dihydropyridine-3,5-dicarbonitrile as the tertiary alkyl radical precursor furnished the desired allenol 58 in good yield and high d.r. & ee (eq. 3). When using the primary radical precursor, only moderate amount of allenol 59a was isolated (>20:1 d.r., 85% ee), accompanied with 28% direct alkylation by-product 59b in 76% ee (eq.

4). These results indicate that the single electron reduction of the primary alkyl radical by $\text{Cr}^{\text{II}}/\text{L}$ could compete with its addition to 1,3-enynes. Accordingly, we included these two examples in Scheme 2, and edited the main text as follows:

“For example, moderate to good yields and high diastereo- and enantioselectivities are achieved for the alkyl radical precursors with various alkyl substituents, such as cyclohexyl, oxacyclohexyl, azacyclohexyl, cyclopentyl, cyclopentenyl, and *tert*-butyl (**52–58**). However, using DHP ester with a primary alkyl substituent furnished the desired allenol **59a** in moderate yield (42%, >20:1 d.r., 85% ee), accompanied by 28% direct alkylation product **59b** in 76% ee. These results indicate that the single electron reduction of the primary alkyl radical by $\text{Cr}^{\text{II}}/\text{L}$ could compete with its addition to 1,3-enynes.”

As shown following, under a slightly modified condition, the NHPI esters could serve as a suitable radical precursor with HE as the reductant, furnishing the desired allenols in moderate yield and good d.r. & ee. We have included 4 examples of NHPI esters as Figure 3b. However, Katritzky salts could only provide a trace amount of the desired allenols under a similar condition. Further optimizations of reductants, solvents, ligands, and photocatalysts gave no improvement. Accordingly, we added the following statement to the main text.

“The past decade has witnessed the breakthroughs in *N*-(acyloxy)phthalimides (NHPI esters) as the redox-active esters in decarboxylative cross-couplings. NHPI esters are stable, readily available from carboxylic acids, and thus serve as efficient alkyl radical precursors. Gratifyingly, NHPI esters also work well under a slight modified condition with Hantzsch ester as the reductant, furnishing the desired allenols in moderate to good yield and high stereoselectivity (Figure 3b, 1, 31, 53, and 58).”

Comment 8: Since the obtained allenol structure is frequently found in natural products, this referee highly recommends the authors to give one example applied in the synthesis of natural product molecules, so as to further enhance the quality of this work.

Response: We really appreciate this suggestion from the referee. For this three-component reaction, the alkyl radical precursors are limited to secondary or tertiary alkyl radicals, which largely impedes the synthetic application in natural products. To solve this drawback and apply it to bioactive molecules synthesis, further development of more general metallophotoredox systems is underway in our laboratory.

Comment 9: For the reaction mechanism, the authors mentioned that “As observed in the scope study, the steric hindrance of the acetylenic substituents of 1,3-enynes might be critical for the high chemoselectivity, which favors the allenylation product formation from the propargyl intermediate D, instead of the allenyl D’.” If so, is there any evidence to support that the present system is more favorable for the formation of propargyl chromium D? And why the direct nucleophilic attack of D to benzaldehyde was not favorable and observed? In addition, is there any theoretical calculation to support this result? Furthermore, please explain how the diastereoselectivity and enantioselectivity of the newly formed vicinal stereocenters was controlled for this transformation.

Response: We thank the referee for these comments.

This statement for regioselectivity control is based on the results in substrate scope exploration as well as the possible Zimmerman-Traxler transition state. Currently, we do not have any direct experimental evidence to support that the formation of propargyl chromium D is more favored than allenyl chromium D'. Attempts to synthesize these Cr^{III}/L1 complexes all failed. We believe that the isomerization between propargyl Cr D and allenyl Cr D' is faster than the subsequent nucleophilic addition to aldehydes. So the regioselectivity might be determined in the nucleophilic carbonyl addition step via a possible Zimmerman-Traxler transition state.

The direct nucleophilic attack of D to benzaldehyde is also possible via a four-membered transition state, which might have a higher energy barrier than the Zimmerman-Traxler transition state. Therefore, we did not discuss this pathway in the main text.

As for the control of d.r. and ee, we still do not have any direct evidence after considerable efforts. Detailed experimental mechanistic studies, as well as DFT calculations, are underway, which is quite time-consuming. Hopefully, we can elucidate the reaction mechanism in detail in the near future.

We have reorganized the mechanism discussion part based on these suggestions and comments.

Comment 10: The HRMS data for (Scheme 1, entry 6) and (Scheme 2, entry 63) need to be retested.

Response: We thank the referee. The HRMS data for these two entries were retested. The new data are shown as follows:

Entry 6: HRMS (ESI) m/z [M – H₂O + H]⁺ calcd for C₂₃H₃₆SiF: 359.2570, found: 359.2566.

Entry 63 (changed to entry 65): HRMS (ESI) m/z [M – H₂O + H]⁺ calcd for C₁₈H₂₅: 241.1956, found: 241.1954.

Comment 11: Please provide the HRMS data for (Scheme 1, entry 17).

Response: The HRMS data for entry 17 was tested and included in the SI.

HRMS (ESI) m/z [M – H₂O + H]⁺ calcd for C₂₅H₃₇SiO: 381.2614, found: 381.2601.

Reviewer 2:

“... The results reported in this paper are both significant and intriguing. This paper is well written in a concise way as a full paper. The supporting information is also in a good shape. Overall, this synthetic aspect, combined with the mechanistic novelty of this reaction, warrants publication of this interesting work in Nature Communications.”

Comment 1: The closest precedent of this work is a radical carbonyl propargylation by Glorius et al. (ref. 41), which uses the same catalyst system but in racemic fashion. Considering the relevance of these works, it should be highlighted in **Figure 1**.

Response: We thank the referee for this suggestion. We have highlighted the Glorius group's work in **Figure 1a**.

Comment 2: The statement “the reaction was sensitive to moisture and air, probably due to the involvement of highly reactive radical intermediates” in page 5 is not accurate. Normally, radical intermediates are not sensitive to moisture and organochromium species is. Therefore, this statement should be modified.

Response: We really appreciate this helpful suggestion. We modified the related main text part as follows:

“These results indicated that the reaction was sensitive to moisture and air, probably due to the involvement of unstable alkyl chromium complexes.”

Comment 3: The substrate scope is somewhat limited. As for the scope of alkyl-DHPs (Scheme 2), only six alkyl-DHPs have been investigated. To understand the generality and utility of this reaction system, the authors should add more examples of alkyl-DHPs.

(a) The scope does not show any primary or tertiary alkyl-DHPs. Were they not considered or are they not compatible in this reaction?

(b) Could secondary alkyl-DHPs bearing unsymmetric alkyl group be converted to the corresponding products diastereoselectively?

Response: We sincerely thank the referee for these helpful suggestions.

The use of 4-(*tert*-butyl)-2,6-dimethyl-1,4-dihydropyridine-3,5-dicarbonitrile as the tertiary alkyl radical precursor furnished the desired allenol **58** in good yield and high d.r. & ee (eq. 3). When using the primary

radical precursor, only moderate amount of allenol **59a** was isolated (>20:1 d.r., 85% ee), accompanied by 28% direct alkylation by-product **59b** in 76% ee (eq. 4). These results indicate that the single electron reduction of the primary alkyl radical by $\text{Cr}^{\text{III}}/\text{L}$ could compete with its addition to 1,3-enynes. Accordingly, we included these two examples in Scheme 2, and edited the main text as follows:

“For example, moderate to good yields and high diastereo- and enantioselectivities are achieved for the alkyl radical precursors with various alkyl substituents, such as cyclohexyl, oxacyclohexyl, azacyclohexyl, cyclopentyl, cyclopentenyl, and *tert*-butyl (**52–58**). However, the use of DHP ester with a primary alkyl substituent furnished the desired allenol **59a** in moderate yield (42%, >20:1 d.r., 85% ee), accompanied by 28% direct alkylation product **59b** in 76% ee. These results indicate that the single electron reduction of the primary alkyl radical by $\text{Cr}^{\text{III}}/\text{L}$ could compete with its addition to 1,3-enynes.”

To further expand the radical precursor scope, we found that the NHPI esters could serve as a suitable radical precursor with HE as the reductant, furnishing the desired allenols in moderate yield and good d.r. & ee. So we included 4 examples of NHPI esters in Figure 3b, and added the following statement into the main text.

“The past decade has witnessed the breakthroughs in *N*-(acyloxy)phthalimides (NHPI esters) as the redox-active esters in decarboxylative cross-couplings. NHPI esters are stable, readily available from carboxylic acids, and thus serve as efficient alkyl radical precursors. Gratifyingly, NHPI esters also work well under a slightly modified condition with Hantzsch ester as the reductant, furnishing the desired allenols in moderate to good yield and high stereoselectivity (Figure 3b, **1**, **31**, **53**, and **58**).”

As shown following, the use of a secondary alkyl-DHP bearing unsymmetric alkyl group could furnish the desired allenol product, but in poor diastereoselectivity (1:1 d.r., **eq. 5**). The result indicates that the chiral catalyst is not involved into the process of alkyl radical addition to 1,3-enynes. However, using the DHP-ester with a benzyl group gave the direct alkylation product in 1:1 d.r. without any desired allenol product (**eq. 6**), which reveals that the radical nature is crucial for this three-component reaction.

Comment 4: Only a few variations from standard conditions are shown in Table 1. As a research paper, reaction condition optimization details should be given in the manuscript or SI, which can help readers better understand the reaction.

Response: We appreciate this valuable suggestion. To provide more information about the effect of reaction parameters, we have tested additional solvents, photocatalyst, and chiral ligands and extended table 1 to 20 entries. The related main text part was edited accordingly. The new Table 1 was shown as follows:

Table 1. Effect of reaction parameters.

entry	variation from "standard conditions"	yield [%]	dr ^b	ee ^c [%]
1	None	>95	20:1	94
2	without CrCl_2	<2	–	–
3	without 4-CzIPN	<2	–	–
4	without blue LED	<2	–	–
5	L2 , instead of $(S,R)\text{-L1}$	>95	20:1	92
6	L3 , instead of $(S,R)\text{-L1}$	84	3.1:1	47
7	L4 , instead of $(S,R)\text{-L1}$	68	1.6:1	19
8	L5 , instead of $(S,R)\text{-L1}$	52	5.5:1	34
9	L6 , instead of $(S,R)\text{-L1}$	71	1.8:1	46
10	L7 , instead of $(S,R)\text{-L1}$	79	2.8:1	93
11	DME, instead of THF	72	10:1	94
12	MeCN, instead of THF	85	20:1	95
13	EtOAc, instead of THF	>95	20:1	94
14	$[\text{Ir}(\text{dF}(\text{CF}_3)\text{ppy})_2(\text{dtbpy})]\text{PF}_6$ instead of 4-CzIPN	>95	11:1	88
15	5 mol% CrCl_2 , 6 mol% $(S,R)\text{-L1}$	74	20:1	94
16	0.1M, instead of 0.05M, in THF	>95	12:1	93
17	1.2, instead of 1.5, equiv 1 and 2	80	20:1	94
18	1.2, instead of 1.5, equiv 1	90	18:1	93
19	1.0 equiv H_2O was added	<2	–	–
20	1 mL air (added via syringe)	55	14:1	87

Comment 5: In Scheme 2, some rr values are given for substrates 65-67. What is the other regioisomer? 1,2-functionalization? This information should be given as a footnote in Scheme 2.

Response: We thank the referee for this helpful suggestion. The other regioisomer refers to the propargylation product from the 1,2-functionalization. We added a footnote to Scheme 2 to clarify the regioselectivity.

^c The yield is for allenol product, and the minor regioisomer refers to the propargylation product from the 1,2-functionalization."

Comment 6: ¹⁹F-NMR spectra of **6** and **13** should be provided.

Response: These two ¹⁹F-NMR spectra were provided in the updated SI.

Reviewer 3:

Comment 1: The first challenge in line 66 should be regioselectivity control of 1,4-functionalization versus 1,2-functionalization, rather than chemoselectivity control. And some challenges should be refined on the basis of ref 45.

Response: We sincerely thank the referee for these helpful suggestions. Accordingly, the “chemoselectivity control” was changed to “regioselectivity control”. And the second challenge statement was removed on the basis of the citation (*Angew. Chem. Int. Ed.* **2022**, *61*, e202117114.). The updated main text is shown below:

“To achieve this goal, several challenges have to be addressed: (1) the regioselectivity control of 1,4-functionalization versus 1,2-functionalization; (2) the proper choice of radical precursors and photocatalysts to maintain the catalytic cycle; (3) the inhibition of quickly occurring side reactions from reactive radical intermediates or organochromium complexes.”

Comment 2: Please provide the ee values in entries 6 and 7, which may provide more information for readers to know the stereo control with L3 and L4.

Response: The ee values were provided in the updated Table 1.

Comment 3: The reason why the reaction was sensitive to moisture, is not attribute to radical intermediate.

Response: We really appreciate this comment. We have changed the related main text as follows:

“These results indicated that the reaction was sensitive to moisture and air, probably due to the involvement of unstable alkyl chromium complexes.”

Comment 4: In the proposed mechanism, the authors indicated that the isopropyl radical could either reversibly add to the low valent Cr^{II}/L1 to generate an off-cycle alkyl Cr^{III}/L1 complex. Did the authors detect any side product? Did the off-cycle reaction affect the titled reaction? How to control the off-cycle reaction?

Response: We thank the referee for these comments. This statement is mainly based on the previous study on alkyl Cr complex (*J. Am. Chem. Soc.* **2010**, *132*, 17325). Experimentally we do observe trace amount of the alkylation by-product from off-cycle alkyl-Cr^{III}/L1 complex under certain cases. Generally, the addition rate to pi-bond of primary alkyl radicals is slower than that of

secondary or tertiary radicals. And primary alkyl Cr complex is typically more stable than the secondary alkyl Cr complex. As shown in equation 4, the use of DHP ester with a primary alkyl substituent could lead to 28% by-product **59b**, generated from the off-cycle alkyl-Cr complex. In conclusion, we believe that the effect of the off-cycle reaction is largely determined by the nature of the alkyl radicals, and the chiral ligand might also matter. For three-component reactions with secondary or tertiary radical precursors, the effect of the off-cycle reaction is slight or neglectable on the reaction yield in most cases.

To provide more information, we included example 59 into Scheme 2 and edited the related main text accordingly.

REVIEWERS' COMMENTS

Reviewer #1 (Remarks to the Author):

The authors have well responded to the issues raised. The present form is suitable for publication.

Reviewer #2 (Remarks to the Author):

I have checked the authors' response to reviewers' comments, as well as the revised manuscript. All the issues raised by the reviewers have fully addressed by the authors and the revised manuscript can be accepted at this stage.

Reviewer #3 (Remarks to the Author):

Authors have well addressed the questions raised by this review, and the revised manuscript was improved significantly. Thus, this referee recommends to publication this manuscript.